# An Innovative Machine Learning Approach to Predict the Dietary Fiber Content of Packaged Foods

**DOI:** 10.3390/nu13093195

**Published:** 2021-09-14

**Authors:** Tazman Davies, Jimmy Chun Yu Louie, Tailane Scapin, Simone Pettigrew, Jason HY Wu, Matti Marklund, Daisy H. Coyle

**Affiliations:** 1The George Institute for Global Health, Faculty of Medicine, University of New South Wales, Sydney, NSW 2042, Australia; jimmyl@hku.hk (J.C.Y.L.); tailane.scapin@deakin.edu.au (T.S.); spettigrew@georgeinstitute.org.au (S.P.); jwu1@georgeinstitute.org.au (J.H.W.); mmarklund@georgeinstitute.org.au (M.M.); dcoyle@georgeinstitute.org.au (D.H.C.); 2School of Biological Science, Faculty of Science, The University of Hong Kong, Hong Kong 999077, China; 3Nutrition in Foodservice Research Centre, Federal University of Santa Catarina, Florianopolis 88040-900, Brazil; 4Department of Epidemiology, John Hopkins Bloomberg School of Public Health, Baltimore, MD 21205, USA; 5Department of Public Health and Caring Sciences, Uppsala University, 75122 Uppsala, Sweden

**Keywords:** dietary fiber, machine learning, computer science, public health

## Abstract

Underconsumption of dietary fiber is prevalent worldwide and is associated with multiple adverse health conditions. Despite the importance of fiber, the labeling of fiber content on packaged foods and beverages is voluntary in most countries, making it challenging for consumers and policy makers to monitor fiber consumption. Here, we developed a machine learning approach for automated and systematic prediction of fiber content using nutrient information commonly available on packaged products. An Australian packaged food dataset with known fiber content information was divided into training (*n* = 8986) and test datasets (*n* = 2455). Utilization of a k-nearest neighbors machine learning algorithm explained a greater proportion of variance in fiber content than an existing manual fiber prediction approach (*R*^2^ = 0.84 vs. *R*^2^ = 0.68). Our findings highlight the opportunity to use machine learning to efficiently predict the fiber content of packaged products on a large scale.

## 1. Introduction

Dietary fiber is the part of plant material that is resistant to digestion and includes polysaccharides, oligosaccharides, and lignins [1]. In the diet, fiber is naturally found in fruits, vegetables, and cereals and grain products, and can also be added to food products in refined form (e.g., inulin). Fiber is an important component of a healthy diet as it promotes fullness and reduces cholesterol and blood sugar levels [2]. The World Health Organization (WHO) recommends that individuals consume a minimum of 25 g of fiber per day to help to protect against disease [3], as intakes lower than this are associated with type 2 diabetes [4], cardiovascular disease [5], and various other health issues [6]. However, underconsumption of fiber is prevalent worldwide [7,8,9,10], including in Australia where the majority of individuals do not meet this target level [11].

Packaged food products tend to be lower in fiber compared to unprocessed or minimally processed foods and higher in unfavorable nutrients such as added sugar, sodium, and saturated fat [12,13,14]. In Australia, nearly two-thirds of the foods and beverages sold by retailers are packaged products, and this proportion appears to be growing [15]. Currently, the labeling of fiber content on the nutrition information panels of packaged products is voluntary in Australia unless a fiber content claim is made, in which case it’s mandatory [16]. As a result, it is difficult for consumers to make informed food purchase decisions and challenging for policy makers to monitor trends in the fiber content of packaged foods. While approaches to predict nutrient composition based on available ingredient and nutrient information have been developed [17,18,19,20], these approaches are resource intensive as they rely on a high degree of manual expert nutritionist input.

A novel approach that could overcome the need for manual fiber prediction would be to use machine learning methods. A major field of artificial intelligence, machine learning can allow a computer system to develop an algorithm that can map input information (e.g., on-pack product details) to a specified output (e.g., fiber content) based on training data [21]. These methods have been applied in many areas including antioxidant protein identification [22], transcriptome subtype classification [23], malnutrition screening [24], and food item recognition [25]. In the context of nutrient prediction, Chin et al. (2019) have demonstrated the feasibility of using gradient boosting to predict the lactose content of dietary recall items given known nutrient information [26]. Additionally, Ma et al. (2021) have shown the possibility of using neural networks to predict the carbohydrate, protein, and sodium content of branded foods based on their ingredients [27]. However, despite the importance of fiber as part of a healthy diet, machine learning techniques have not yet been applied to the prediction of fiber content in packaged foods and beverages.

This study therefore aimed to (i) develop a machine learning approach for the prediction of fiber content in packaged foods and beverages based on commonly available nutrient information and (ii) apply this algorithm to a selection of packaged products that do not report fiber content to comprehensively estimate fiber levels in the Australian packaged food supply for the first time.

## 2. Materials and Methods

### 2.1. Dataset

The George Institute’s Australian FoodSwitch database [28] was used to conduct analyses. This database contains product data on more than 80,000 packaged foods and beverages that have been sold in Australia since January 2013. Most of the data (~60%) are captured by trained data collectors during annual in-store surveys at four supermarket stores in metropolitan Sydney, one from each of the four largest supermarket chains (Coles, Woolworths, IGA, Aldi). Using a bespoke smartphone application, a trained data collection team captures images of each product’s packaging and then extracts all key information from these images, such as barcode, brand name, product name, nutrients, and ingredients. The database also contains data that are crowdsourced using the FoodSwitch smartphone application (~30%) and data provided directly by the food industry (~10%). All product data undergo quality checks. Products are classified into 67 food and beverage categories consistent with the Global Food Monitoring Group categorization system (e.g., ‘biscuits’, ‘bread’, and, ‘cakes, muffins, and pastries’) [29] (Appendix A).

### 2.2. Exclusion Criteria and Data Preparation

In total, 79,656 foods and beverages with unique barcodes spanning January 2013 to January 2020 were extracted from FoodSwitch. Of these, we excluded products that did not contain a nutrition information panel (*n* = 10,683), products that did not contain an ingredients list (*n* = 1151), products that reported an unclean ingredients list (e.g., the number of opening parentheses was not equal to the number of closing parentheses) (*n* = 9948), products that were a pack size variant of an included product (*n* = 5002), and products that did not contain brand information (*n* = 93). Additionally, we excluded food and beverage categories that either (i) contributed less than 0.8% of fiber intake in Australia [30] (e.g., ‘cheese’, ‘cooking oils’, and ‘honey’), or (ii) had less than 25% of products reporting fiber content (e.g., ‘processed meat’) (*n* = 41,338) (Appendix A). The above exclusions resulted in 21,246 products, spanning 14 different food and beverage categories, that were available for analysis.

Included products that reported fiber content were organized into brand name groups (e.g., ‘Pringles Original’, ‘Pringles BBQ’, and ‘Pringles Salt and Vinegar’ were organized into a ‘Pringles’ brand group). Then, 75% of brands and all their products were randomly allocated to the training dataset (*n* = 8986) and 25% of brands and all their products were allocated to the test dataset (*n* = 2455). The training dataset was used for algorithm development and the test dataset was used to evaluate the predictive algorithm. The remaining products that did not report fiber content (*n* = 9805) were eligible for subsequent fiber content prediction (Figure 1).

### 2.3. Product Representation

All products were represented using six nutrient features that were obtained or derived from the nutrition information panel: total sugar (g per 100 g or 100 mL), starch (g per 100 g or 100 mL), protein (g per 100 g or 100 mL), saturated fat (g per 100 g or 100 mL), unsaturated fat (g per 100 g or 100 mL), and sodium (mg per 100 g or 100 mL). All chosen nutrients were mutually exclusive to minimize any multicollinearity in the algorithm. Starch content was determined by subtracting total sugar from carbohydrates, and unsaturated fat content was determined by subtracting saturated fat from total fat. Definitions of all nutrients are provided by Food Standards Australia New Zealand [16]. To allow the algorithm to place a similar emphasis on nutrients of different ranges, all nutrients were normalized by dividing by minimum and maximum nutrient values in the training dataset (i.e., $X’=\frac{{X\;-\;X}_{\min}}{X_{\max}{\;-\;X}_{\min}}$) (Appendix A). Food category was dummy-coded.

### 2.4. Predictive Algorithm

We chose to utilize k-nearest neighbors (KNN) for fiber content prediction. Prior studies have shown that KNN is apt at predicting nutritional phenotypes [31] and childhood obesity [32] based on nutrient variables, which suggests the approach might also be apt at predicting fiber content based on nutrient variables. Additionally, the internal structure of KNN is simple and aids in interpretation [33].

The KNN algorithm first calculated the Manhattan distance (d) between a query product and each training product by summing the absolute differences between each pair of normalized nutrient values (Equation (1)),
(1)$$d(x,\;y)=\overset{6}{\underset{i=1}{\sum }}{|x}_{i}{-y}_{i}|$$
where x_i_ and y_i_ refer to the *i*-th normalized nutrient feature of the query point and training point, respectively.

A product’s fiber content (Q) was then predicted by calculating an inverse distance weighted average of the eight nearest product’s fiber values (Equation (2)),
(2)$$Q(q,\;w)=\;\frac{\;\sum_{j=1}^{8}{\;w}_{j}q_{j}\;}{\sum_{j=1}^{8}{\;w}_{j}}$$
where w_j_ is the weight of the *j*-th closest product (calculated using 1/d_j_) and q_j_ is the fiber content of the *j*-th closest product.

Dummy-coded food categories ensured nearest neighbors were always selected from the same food category as the query product. A visual presentation of the KNN fiber prediction algorithm is displayed in Figure 2. An example application of the KNN algorithm is provided in Appendix A.

The utilized hyperparameters (i.e., Manhattan distance, inverse weight function, and eight neighbors) were chosen as they exhibited one of the highest five-fold cross-validated *R*^2^ scores on the training dataset (Appendix A). The Manhattan distance metric and the Euclidean distance metric performed similarly well, but we chose the Manhattan distance metric due to the ease of calculation. The inverse weight function, which placed a stronger emphasis on nearby neighbors than distant neighbors, consistently outperformed uniform weighting of neighbors. Lastly, eight neighbors showed a plateau in algorithmic performance and was therefore chosen.

### 2.5. Statistical Analyses

Python 3.7 and the libraries *scikit-learn*, *numpy, pandas*, and *seaborn* were used to conduct data analyses. To assess the algorithm’s performance on the test dataset, predicted values were compared against the values reported on the packaging. We calculated the coefficient of determination (*R*^2^), mean absolute error (MAE), and Spearman rank correlation (*ρ*). These metrics were calculated overall, by food and beverage category, and by quartile of all first neighbor distances (i.e., all d_1_ values).

After applying the KNN fiber prediction algorithm to the test dataset, we assessed the algorithm’s ability to identify products that contained negligible fiber density (<0.9 g per 100 g or 100 mL), low fiber density (0.9–3.7 g per 100 g or 100 mL), medium fiber density (3.7–7.3 g per 100 g or 100 mL), and high fiber density (>7.3 g per 100 g or 100 mL). These cut-offs were based on the nutrient profiling scoring criteria [34] that underpin the Australian front-of-pack Health Star Rating system and correspond to 0, 1–3, 4–7, and 8–15 fiber points, respectively. We then calculated classification accuracy (CA), precision, and recall scores.

We also applied the manual recipe-based nutrient prediction approach developed by Ng et al. (2015) [17] to the test dataset, which is currently the only published nutrient prediction approach that can be used to predict fiber. In brief, this approach involved (i) disaggregating a product’s ingredients list and manually mapping each ingredient to an ingredient in a nutrient composition database as previously described [35,36], (ii) predicting the proportion of each ingredient in the product using a mathematical optimization algorithm (which is necessary as ingredient proportions are typically not reported on the packaging), and (iii) summing the amount of fiber in each ingredient to determine overall fiber content (Appendix A). Steps (ii) and (iii) were applied by T.D. The validity of this approach was assessed using the same performance metrics to benchmark the KNN fiber prediction algorithm 

After applying the KNN fiber prediction algorithm to products that did not report fiber content, we calculated median (interquartile range) fiber values for all included products. Calculations were conducted overall and by food category. To estimate the differences in fiber levels between products that do and do not report fiber content, differences between all reported and all non-reported (but predicted) fiber values were compared using a Wilcoxon signed-rank test, where statistical significance was considered using *p* < 0.05.

## 3. Results

### 3.1. Performance of the KNN Fiber Prediction Algorithm

The performance of the KNN algorithm on the testing dataset was visualized in a scatter plot (Figure 3). Overall, the KNN algorithm explained a large proportion of variation in fiber content (*R*^2^ = 0.84). However, some very large errors were present; 5% (*n* = 123) predictions differed from the reported value by more than 5 g per 100 g or 100 mL. Examination of these products showed that most (*n* = 80) utilized a highly distant nearest neighbor for fiber content prediction (i.e., *d*_1_ was in the fourth quartile of all *d*_1_ values).

At the category level, the algorithm was most apt at explaining variance in fiber content for ‘other cereal and grain products’ (*R*^2^ = 0.95) and ‘soup’ (*R*^2^ = 0.89), while least apt at explaining variation in fiber content for ‘pasta’ (*R*^2^ = −0.55) (Figure 4). When three highly erroneous predictions in the ‘pasta’ category were excluded (i.e., spaghetti, fusilli, and tufoli products each with a reported fiber value of 2.1 g and predicted fiber value of 8.5 g per 100 g), the performance assessment noticeably increased (*R*^2^ = 0.23). A strong predictor of the accuracy of the algorithm was the proximity of the nearest neighbor, as products with a close nearest neighbor exhibited excellent accuracy (*d*_1_: 0.000–0.020, *R*^2^ =0.99, MAE = 0.49 g per 100 g or 100 mL, *ρ* = 0.95, *n* = 614/2455) while products with a distant nearest neighbor exhibited much less accurate predictions (*d*_1_: 0.080–0.675, *R*^2^ = 0.55, MAE = 2.66 g per 100 g or 100 mL, *ρ* = 0.64, *n* = 614/2455).

After fiber predictions were grouped into fiber density classes, the classification accuracy of the KNN algorithm was calculated to be 71%. The algorithm was most apt at identifying products that had high fiber density (precision: 0.76, recall: 0.81), while worst at identifying products that had negligible fiber density (precision: 0.61, recall: 0.56) (Appendix A).

Benchmarking against the manual recipe-based nutrient prediction approach developed by Ng et al. (2015) [17] showed that, across all test products, the KNN algorithm explained a greater proportion of variance in fiber content (*R*^2^: 0.84 vs. 0.68), produced predictions with smaller absolute deviations from reported values (MAE: 1.52 g vs. 2.07 g per 100 g or 100 mL), more accurately ranked products from lowest to highest in fiber content (*ρ*: 0.84 vs. 0.77), and more accurately predicted fiber density class (CA: 71% vs. 62%). Of the 14 food and beverage categories, there were only four for which the manual approach was superior across a majority of the performance metrics: ‘breakfast cereals’, ‘cakes, muffins, and pastries’, ‘nuts and seeds’, and ‘ready meals’ (Appendix A).

### 3.2. Fiber in the Australian Packaged Food Supply

Overall, 54% of included products reported fiber content, with ‘cereal and nut-based bars’ most frequently reporting fiber content (90%) and ‘cakes, muffins, and pastries’ least frequently reporting fiber content (29%). The median fiber content also varied by food category; ‘breakfast cereals’ reported the highest median fiber value (8.8 g per 100 g) and ‘fruit and vegetable juices’ the lowest (0.6 g per 100 mL). Products that reported fiber content had significantly higher fiber content than products that did not (median: 3.8 g vs. 2.8 g per 100 g or 100 mL, *p* < 0.001) (Table 1).

## 4. Discussion

This is the first study to use machine learning to predict the fiber content of packaged products. We utilized a KNN algorithm that predicted a product’s fiber content by considering the fiber values of the eight most similar products in the same food category (i.e., closest neighbors in feature space). While fiber content labeling remains voluntary in Australia and approximately half of all packaged foods report fiber content, this algorithm can be used to predict fiber content for 14 categories that account for the majority of fiber intake.

A strong predictor of the accuracy of the KNN algorithm is the proximity of the nearest neighbor, as products with a close nearest neighbor exhibited the highest accuracy while products with a distant nearest neighbor exhibited modest accuracy. This is probably because closer products tend to have more similar ingredient profiles and consequently more similar fiber content. Therefore, we advise that all predictions that utilize a distant first neighbor (d_1_ > 0.08) are interpreted with caution. As additional product data are included in FoodSwitch and the size of the training dataset increases, we expect the nearest neighbors to be closer and the predictive accuracy of the KNN fiber prediction algorithm to increase.

According to our results, the KNN algorithm offers substantially higher accuracy than Ng et al.’s [17] manual recipe-based nutrient prediction approach. Unlike Ng et al.’s approach, the KNN algorithm does not rely on the prediction of ingredient proportions as an intermediate step, which may be the reason for its superior performance. If ingredient proportions were known and utilized in Ng et al.’s approach, subsequent ingredient fiber amount calculations may be more accurate and Ng et al.’s approach might outperform the KNN algorithm. Either way, the KNN algorithm is more efficient than Ng et al.’s approach as it can be used for automated and systematic prediction of fiber content on a very large scale (provided the required input information is available). Additionally, as the algorithm always predict the same fiber content given the same input information, the approach also minimizes inter-rater reliability issues inherent in the subjectivity of Ng et al.’s approach.

Compared to the gradient boosting lactose prediction algorithm developed by Chin et al. [26], the KNN fiber prediction algorithm in this study appears to have higher predictive performance (*R*^2^: 0.33 vs. 0.84, *ρ*: 0.75 vs. 0.84), with this difference likely being due to the relatively small training dataset in Chin et al.’s study (*n* = 378). Interestingly, while Ma et al.’s neural network carbohydrate, protein, and sodium prediction algorithms were developed using a training dataset much larger than the one used in this current study (*n* = 212,178 vs. *n* = 8986) [27], the performance metrics are within a similar range as the KNN fiber prediction algorithm (*R*^2^: 0.90, 0.80, and 0.70, respectively, vs. *R*^2^: 0.84). This suggests KNN could have similar or even greater predictive capacity than neural networks in a nutrient prediction context. However, comparisons with other studies must be made with caution given the different data and target nutrients.

The KNN approach is also intrinsically interpretable as neighboring products can be retrieved to explain a given prediction. This is in contrast to Chin et al. [26] and Ma et al.’s [27] approaches that rely on frameworks that are often described as “black boxes” and that can only be understood in terms of post hoc interpretability techniques [33]. With this degree of interpretability, it is likely the KNN approach will be more easily trusted and more readily deployed [33]. The KNN approach can also be easily adapted to predict other nutrients of public health interest that are often omitted on packaged products (e.g., trans fat), albeit the predictive accuracy for these nutrients is unknown.

A key finding of this study is that packaged products that omit fiber information are skewed towards being low in fiber. Consequently, consumers may be less apt at identifying products that are low in fiber. To address this, KNN-based fiber predictions could be integrated into novel barcode scanning phone applications (e.g., FoodSwitch [28] and MyFitnessPal [37]) to allow consumers to understand the fiber content of their food purchases and select products higher in fiber. This would be particularly helpful for individuals with pre-diabetes and type 2 diabetes, as it is strongly recommended that these individuals consume foods high in fiber while staying within energy intake recommendations [38]. Furthermore, policy makers and researchers could benefit from these predictions through being able to assess and monitor fiber levels in the packaged food supply, which may be important in developing nutrition policies.

This study has a number of strengths. The accuracy of the KNN algorithm was assessed at the category level using multiple performance metrics, consequently allowing us to estimate algorithmic performance in a variety of contexts. The use of benchmarking also allowed us to confirm the algorithm was generally more accurate than an existing manual nutrient prediction approach, as well as understand the additional degree of accuracy offered by the KNN algorithm. Additionally, the use of a test dataset containing real products allowed us to estimate how the approach would perform in practice. This is also the first study to predict and examine the fiber composition of the Australian packaged food supply.

There are also several limitations of this study. First, as only products that reported fiber content could be used during algorithm development, there was a level of bias in the training and test datasets, with both likely skewed towards being high in fiber content. Second, excluding products without a clean ingredients list removed a considerable portion of products (11%), and it was likely this was a more pronounced issue for products with more complex ingredient lists. Third, this paper did not develop and compare different machine learning fiber prediction algorithms. However, based on comparison with other machine learning nutrient prediction studies, we believe other algorithms would perform similarly or worse than KNN. Lasty, given our algorithm was developed and evaluated using Australian product data, the generalizability of the KNN prediction algorithm to the food supply in other countries is unknown. Different countries have different nutrient labelling conventions for packaged foods (e.g., available carbohydrates is displayed in Australia [16] while total carbohydrates is displayed in the US [39]) and the latter might hinder the algorithm’s selection of neighbors and subsequent fiber prediction. As this study did not assess the performance of the algorithm on external validation datasets, future studies are needed to investigate the accuracy of the KNN fiber prediction approach when applied to datasets from other countries.

## 5. Conclusions

KNN can be used to predict the fiber content of Australian packaged foods and beverages based on commonly available nutrient information. The algorithm is more accurate than an existing manual nutrient prediction approach and can automate fiber content prediction on a large scale. In comparison to previous neural network nutrient prediction approaches, the KNN fiber algorithm offers similar predictive performance with greater interpretability. These predictions can be used to monitor fiber in the Australian packaged food supply, as well as inform interventions aimed at increasing fiber intake. At this stage, the applicability of the KNN fiber prediction algorithm to countries with different packaged food systems is unknown. However, other countries can consider developing a KNN fiber prediction algorithm using an in-country packaged food dataset and the same methodology used in this paper.

## Figures and Tables

**Figure 1 nutrients-13-03195-f001:**
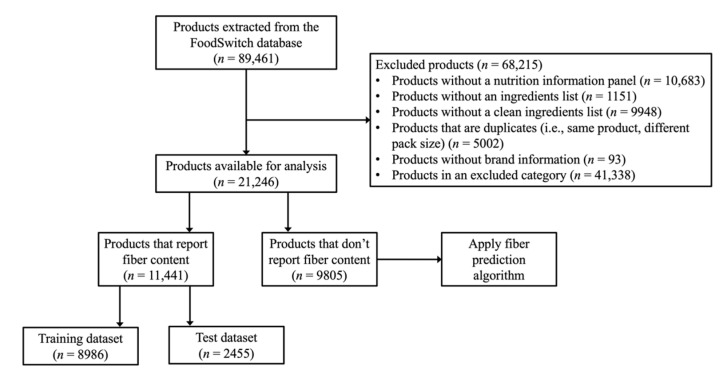
Exclusion criteria and data preparation flow chart.

**Figure 2 nutrients-13-03195-f002:**
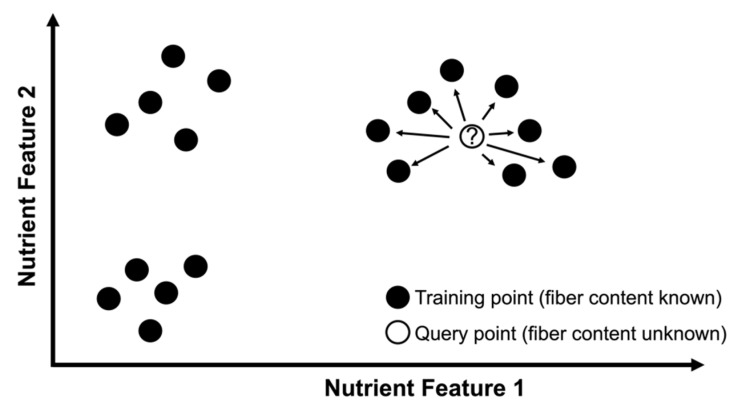
Visual presentation of k-nearest neighbors (KNN) algorithm for fiber content prediction. A query product’s fiber content is predicted by considering the eight neighboring products’ fiber values in feature space. Demonstration is in 2D feature space for easier visualization, e.g., nutrient feature 1 could be normalized total sugar content per 100 g or 100 mL and nutrient feature 2 could be normalized starch content per 100 g or 100 mL. For the implemented algorithm, the feature space is composed of six normalized nutrient features: total sugar, starch, protein, saturated fat, unsaturated fat, and sodium. Nearest neighbors are always selected from the same food category as the query product.

**Figure 3 nutrients-13-03195-f003:**
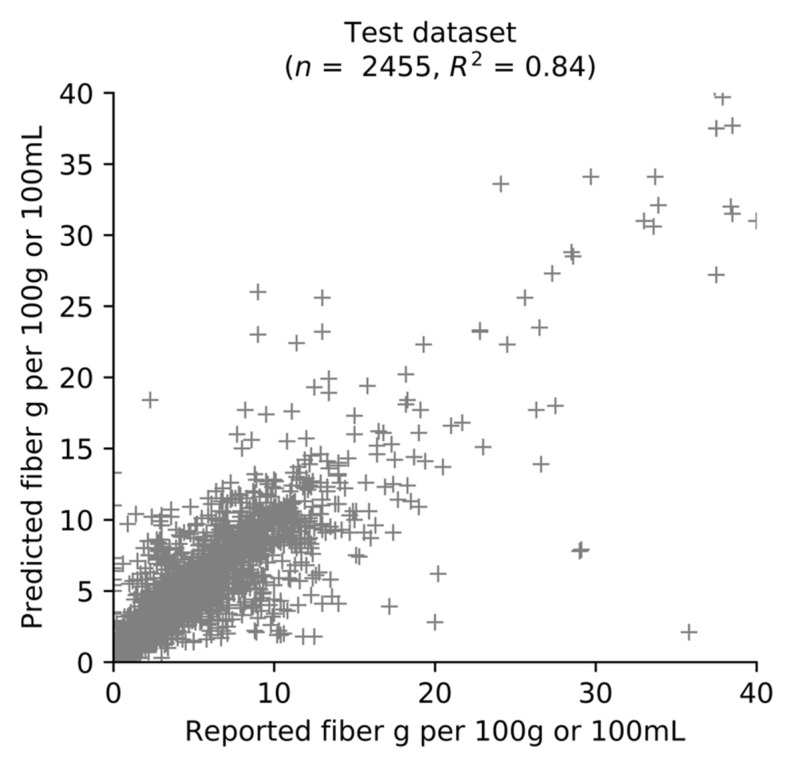
Scatter plot of predicted and reported fiber values on the test dataset using the KNN algorithm. *R*^2^ is the coefficient of determination. It represents the proportion of variance in added sugar content explained by the algorithm, where 1 indicates a perfect fit. For presentation purposes, twelve products were excluded (predicted or reported fiber content >40 g per 100 g or 100 mL).

**Figure 4 nutrients-13-03195-f004:**
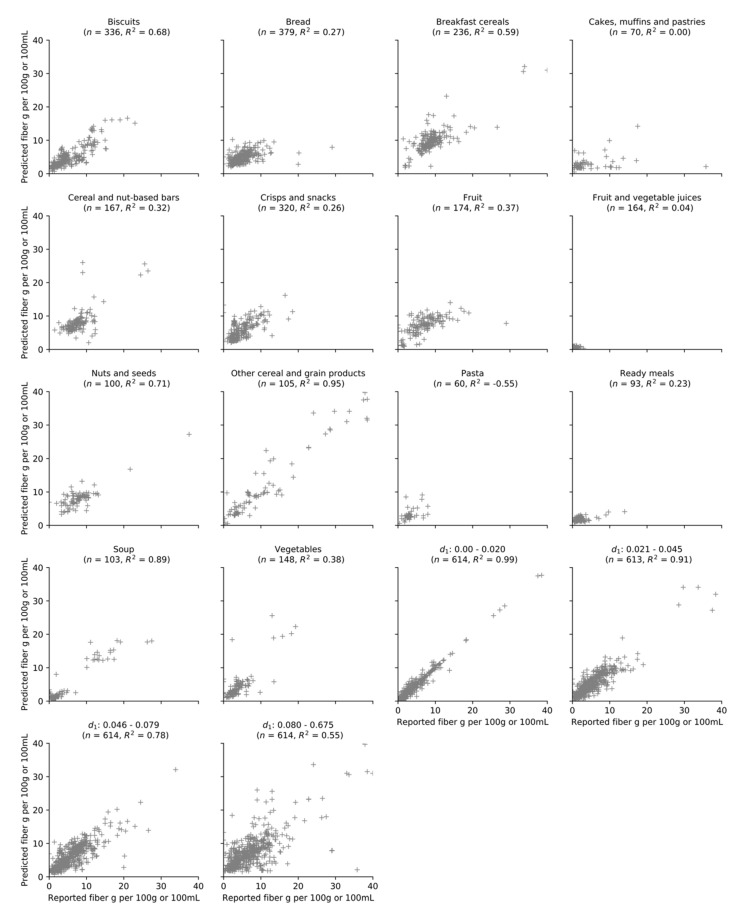
Scatter plots of predicted and reported fiber values on the test dataset using the KNN algorithm, stratified by food category and proximity of the nearest neighbor (d_1_). d_1_ is calculated using $d(x,\;y)=\overset{6}{\underset{i=1}{\sum }}{|x}_{i}{-y}_{i}|$, where x_i_ and y_i_ refer to the *i*-th normalized nutrient value of the query point and nearest neighbor, respectively. *R*^2^ is the coefficient of determination. It represents the proportion of variance in fiber content explained by the algorithm, where 1 indicates a perfect fit. It can take a negative value as it is calculated using the formula: R^2^ = 1 – (sum of squared residuals/total sum of squares). For presentation purposes, twelve products were excluded (predicted or reported fiber content >40 g per 100 g or 100 mL).

**Table 1 nutrients-13-03195-t001:** Fiber in the Australian packaged food supply.

Food Category	Products that Report Fiber Content		Products that Do Not Report Fiber Content		All Products	
	*n* (%)	Median (IQR ^1^) Reported Fiber g per 100 g or 100 mL	*n*	Median (IQR) Predicted Fiber g per 100 g or 100 mL	*n*	Median (IQR) Fiber g per 100 g or 100 mL
Bread	1279 (64)	4.3 (3.1)	728	3.8 (1.6)	2007	4.0 (2.7)
Biscuits	978 (40)	3.5 (3.4)	1468	3.0 (1.6)	2446	3.2 (2.1)
Crisps and snacks	977 (54)	4.0 (4.7)	848	3.7 (2.9)	1825	3.8 (3.8)
Breakfast cereals	1250 (88)	8.8 (4.2)	177	8.7 (4.4)	1427	8.8 (3.2)
Fruit	1126 (63)	5.0 (6.6)	670	5.1 (6.0)	1796	5.0 (6.4)
Cereal and nut-based bars	648 (90)	7.2 (3.5)	76	6.7 (1.6)	724	7.1 (3.3)
Fruit and vegetable juices	818 (50)	0.7 (0.8)	829	0.6 (0.7)	1647	0.6 (0.7)
Vegetables	1127 (49)	3.1 (2.5)	1153	2.5 (1.1)	2280	2.7 (1.9)
Other cereal and grain products	532 (64)	8.5 (14.5)	302	5.8 (6.9)	834	7.6 (9.9)
Soup	505 (52)	1.2 (1.0)	468	1.2 (0.8)	973	1.2 (1.0)
Nuts and seeds	600 (48)	8.0 (3.5)	662	7.2 (3.4)	1262	7.7 (3.2)
Ready meals	574 (40)	1.6 (1.2)	856	1.8 (0.6)	1430	1.7 (0.7)
Cakes, muffins, and pastries	461 (29)	2.2 (2.0)	1120	2.2 (1.1)	1581	2.2 (1.3)
Pasta	566 (56)	3.0 (1.9)	448	3.0 (1.3)	1014	3.0 (1.6)
Total	11,441 (54)	3.8 (5.5)	9805	2.8 (2.8)	21,246	3.2 (4.4)

^1^ IQR is interquartile range.

## Data Availability

The data presented in this study are not publicly available due to the proprietary nature.

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
