# Peer review of "An Innovative Machine Learning Approach to Predict the Dietary Fiber Content of Packaged Foods"

_nutrients, 2021, doi:10.3390/nu13093195_

Round 1

Reviewer 1 Report

An interesting study to use machine learning to predict the fiber content of packaged products. The study is of interest for the scientific community,but overall, the manuscript's needs to be thoroughly revised.

The “Introduction” section of the manuscript requires extensive revision. First of all, the authors need to expand the review of literature that is relevant to their study.
Second, and probably most importantly, the aim of the study needs to be properly highlighted and justified. Instead of setting their aim in the frame of a simplistic question (reviewer’s personal point of view), I would suggest that the authors attempt to present the key objectives of their study with regards to what is currently known (i.e. literature), thus highlighting the added value of the paper.

Discussion of results significantly lacks quality.

Conclusions are not presented properly. Mainly, this is because the presentation of results lacks any discussion in terms of the international literature.

Author Response

Hello reviewer 1. Please find responses attached.

Reviewer 2 Report

In this study, Davies et al. proposed a machine learning model to predict the dietary fiber content of packaged foods. The dataset is collected from the Australian packaged food dataset and the authors created a kNN model to work on this data. Even the idea looks interesting, there are some major points that need to be addressed:

1. In the "Introduction", there must have more literature review on machine learning-based prediction model for package foods.

2. Why did the authors select kNN as their machine learning algorithm? The authors should have a comparison among different algorithms to see that their choice is an optimal one. Also, the hyperparameters should be tuned to ensure a fair comparison.

3. The authors should describe more the features/variables of the model.

4. A big concern is that whether this model can be applied to predict other datasets related to food packages.

5. The authors should have some external validation datasets.

6. The authors should compare the predictive performance to previously published works on the same problem/data.

7. Machine learning is well-known and has been used in previous studies i.e., PMID: 33036150 and PMID: 33735760. Thus, the authors are suggested to refer to more works in this description to attract a broader readership.

8. Quality of figures should be improved, i.e., Figure 4.

9. With this amount of data, the authors should discuss some potentials of using deep learning to extract the features as well as perform the prediction for this problem.

Author Response

Hello Reviewer 2. Please find the responses attached.

Round 2

Reviewer 1 Report

The authors have been improved significantly the manuscript. However, revise the conclusions part please.

Author Response

Thank you for your feedback.

We have revised the conclusion to have more explicit comparison to international literature pertaining to nutrient prediction approaches. We have also highlighted how the applicability of the algorithm to international datasets is unknown.

KNN can be used to predict the fiber content of Australian packaged foods and beverages based on commonly available nutrient information. The algorithm is more accurate than an existing manual nutrient prediction approach and can automate fiber content prediction on a large scale. In comparison to previous neural network nutrient prediction approaches, the KNN fiber algorithm offers similar predictive performance with greater interpretability. These predictions can be used to monitor fiber in the Australian packaged food supply, as well as inform interventions aimed at increasing fiber intake. At this stage, the applicability of the KNN fiber prediction algorithm to countries with different packaged food systems is unknown. However, other countries can consider developing KNN fiber prediction algorithm using an in-country packaged food dataset and the same methodology used in this paper.

We hope this addresses your queries regarding the conclusion. Please let us know if you'd advise further changes.

Best,

The Authors

Reviewer 2 Report

My previous comments have been addressed well.

Author Response

Thank you.